# Weak relationship between remotely detected crevasses and inferred ice rheological parameters on Antarctic ice shelves

Cristina Gerli[1], Sebastian Rosier[1,2], G. Hilmar Gudmundsson[1], Sainan Sun[1]

[1] Department of Geography and Environmental Sciences, Northumbria University, Newcastle Upon Tyne, UK
[2] Department of Geography, University of Zurich, Zurich, Switzerland

*Correspondence to*: Cristina Gerli (cristina.gerli@northumbria.ac.uk)

**Abstract.**

Over the past decade, a wealth of research has been devoted to the detection of crevasses in glaciers and ice sheets via remote sensing and machine learning techniques. It is often argued that remotely sensed damage maps can function as early-warning signals for shifts in ice shelf conditions from intact to damaged states and can serve as an important tool for ice sheet modellers to improve future sea-level rise predictions. Here, we provide evidence for Filchner-Ronne and Pine Island ice shelves that remotely sensed damage maps are only weakly related to the ice rate factor field $A$ derived by an ice-flow model when inverting for surface velocities. This technique is a common procedure in ice flow models, as it guarantees that any inferred changes in $A$ relate to changes in ice flow measured through observations. The weak relationship found is improved when investigating heavily damaged shear margins, as observed on Pine Island Ice Shelf; yet, even in this setting, this association remains modest. Our findings suggest that many features identified as damage through remote sensing methods are not of direct relevance to present-day ice-shelf flow. While damage can clearly play an important role in ice-shelf processes and thus be relevant for ice-sheet behaviour and sea-level rise projections, our results imply that mapping ice damage directly from satellite observations may not directly help improve the representation of these processes in ice-flow models.

## 1 Introduction

The advent of high-resolution satellite imagery and the development of new sophisticated algorithms to analyse these products have motivated many studies that track surface crevasse features across the Antarctic continent (Lai et al., 2020; Lhermitte et al., 2020; Zhao et al., 2022; Izeboud and Lhermitte, 2023; Surawy-Stepney et al., 2023). Some of these studies have highlighted the value of remotely sensed crevasses maps for ice sheet modellers when investigating future ice shelf stability (Zhao et al., 2022), initialising or improving creep damage models (Izeboud and Lhermitte, 2023), and evaluating the impact of ice shelf crevasses on ice dynamics ( Lai et al., 2020, Zhao et al., 2022). Specifically, a recent study from Surawy-Stepney et al., (2023), has suggested potential applications of damage fields extracted from satellite imagery as fracture data — remotely sensed maps of fractures — can be used to constrain inverse problems in ice-flow models aiming to infer the effective ice viscosity from surface velocities.

For alpine glaciers, ice caps and ice sheets, where most of the forward motion is either due to ice deformation concentrated close to the bed or caused by basal sliding, surface crevasses are unlikely to have any significant impact except for a possible modification in the effective ice density of the surface layers in which they are confined. For ice shelves, where the ice deformation is generally characterised by uniform horizontal divergence and convergence across the whole ice column, the impact of surface crevasses may not be this easily discounted (Weertman, 1983; Van Der Veen, 1998; Larour et al., 2005; van der Veen, 2007; Khazendar et al., 2009; Colgan et al., 2016). However, only those surface features that actively influence ice flow are pertinent to modelling studies of ice-flow dynamics. In fact, some surface morphological features detected by remote sensing techniques may not necessarily represent crevasses but rather features that can be easily mistaken for crevasses, such as flow stripes (Luckman et al., 2012) or surface expressions of basal fractures (McGrath et al., 2012). In other cases, some features may indeed be crevasses but may have such shallow characteristics that they have limited relevance to ice-flow dynamics or could have transitioned from active to passive states when being advected in regions of compression (Colgan et al., 2016). Hence, it is not a priori clear if the surface morphological features identified through remote sensing products have any direct correspondence to the ice rheological parameters as used in ice-sheet modelling.

Historically, signs of damage concealed beneath undamaged firn in the margins of ice streams have played a crucial role in past changes in ice flow conditions, i.e., the shut-down of Ice stream C in Antarctica (Robin, 1975; Shabtaie, S., & Bentley, 1987; Retzlaff, R., & Bentley, 1993; Smith, B. E., Lord, N. E., & Bentley, 2002). In the modelling community, "ice damage" is generally introduced as a scalar variable that impacts the effective value of a rheological parameter, $A$ — also known as the pre-factor in Glen's flow law (refer to the Methods, Section 2) — by making it rheologically weaker than can be expected for temperate ice (Pralong and Funk, 2005; Borstad et al., 2013; Krug et al., 2014; Sun et al., 2017). Rather than prescribing $A$ as an input field, or alternatively by calculating $A$ from modelled englacial temperatures, many modern ice-flow models infer the $A$ distribution by performing a model inversion using surface velocities and rates of thickness change, sometimes referred to as 'data assimilation'. In contrast to the forward problem, where ice flow is solved as a function of an initial ice sheet state, the inverse problem aims to infer the ice sheet's state (the spatial distribution of $A$) that gave rise to these observations. Solving the inverse problem can be more challenging than its forward counterpart since, in order for it to be well-posed, it requires the introduction of regularisation parameters — described below in the Methods. Nevertheless, the inversion method remains a powerful tool as it ensures that an ice-sheet model's initial state is consistent with the observations available, e.g., surface velocities. Thus, we can employ inverse methods to robustly identify the spatial distribution of $A$ required to generate the specific velocity field observed.

The question now arises if the inferred distribution of $A$ of an ice-flow model agrees with the crevasse maps obtained from satellite imagery. Here, we investigate this association for the whole of Filchner-Ronne Ice Shelf, in the Weddell Sea, as well as Pine Island Ice Shelf, in the Amundsen Sea Embayment. We solve inverse problems at a spatial resolution comparable to

those provided by available crevasse maps. This requires a resolution of about 100 metres, which is considerably higher resolution than typically applied in the study of ice shelves, and a pan-Antarctic inversion at this resolution is currently not feasible. Additionally, these two ice shelves exemplify two contrasting situations. Over the past 20 years, Pine Island Ice Shelf experienced rapid changes (Rignot et al., 2019; Shepherd et al., 2019), transitioning from a state of no crevasses in 1997 to exhibiting extensive crevassed-damaged areas near the grounding line and shear margins in 2019 (Lhermitte et al., 2020). In contrast, the Filchner-Ronne Ice Shelf has remained relatively unchanged in recent decades, primarily due to its location in a colder embayment with less exposure to warm ocean currents (Gardner et al., 2018; Rignot et al., 2019; Shepherd et al., 2019).

To evaluate to what extent maps of crevassing correspond to areas identified as damaged via our model inversion, we treat this as a classification problem. Classification analyses offer a more comprehensive assessment of model performance compared to correlation coefficients, since they can handle imbalanced datasets better and provide insights into the model's performance across different threshold levels. The crevasse products obtained by remote sensing/machine learning techniques represent the true observations to be classified (that is, to match). The predictor variable is the variable used to make a prediction. We treat our model inverted ice rate factor as a predictor for damage and quantify how often it corresponds to crevassed areas (true positives) as against to how often crevasses are incorrectly predicted from areas of damage (false positive). We find, based on this measure, for the regions considered, that the predictive performance is about as good as a random classifier.

## 2 Methods

This investigation comprised three steps. First, we estimated the ice rate factor $A$ by performing a surface velocity inversion using the adjoint capabilities of the numerical ice-flow model, Úa (Gudmundsson, 2013; Gudmundsson et al., 2012). This procedure guarantees that any inferred changes in $A$ relate to changes in the ice flow measured through observations and is a common first-step in many ice-sheet modelling studies (MacAyeal, 1992, 1993; Rommelaere and MacAyeal, 1997; Larour et al., 2005; Raymond and Gudmundsson, 2009; Arthern and Gudmundsson, 2010; Morlighem et al., 2010, 2013; Petra et al., 2012; Gillet-Chaulet, 2020; Barnes et al., 2021). The second step of this analysis consisted in obtaining a crevasse map from remote sensing and machine learning techniques which we compare to the inverted $A$ field. In this study, we used two crevasse datasets: (1) the recently proposed NormalisEd Radon transform Damage detection (NeRD, Izeboud and Lhermitte, 2023) method, which maps areas of surface structural damage on ice shelves using multi-source satellite imagery through a feature contrast approach, and (2) the Antarctic wide crevasse map produced by Lai et al., (2020), which was trained and applied through a Convolutional Neural Network (CNN). Finally, for the third step, we used a standard classification analysis to test and evaluate the relationship between these two observational and modelled products.

For the first step, modelling simulations were performed with the finite element ice flow model, Úa (Gudmundsson, 2013; Gudmundsson et al., 2012), which solves the vertically integrated shallow shelf approximation (SSA) of Macayeal, (1989). Viscous ice deformation was described by Glen's flow law, $\acute{\epsilon} = A\tau^n$, where $A$ is the ice rate factor, $\acute{\epsilon}$ is the effective strain,

$\tau$ the effective stress, and $n$ is the standard constant stress exponent set to 3, while the basal motion for the grounded ice was modelled using a Weertman sliding law $v_b = C\tau_b{}^m$ where $v_b$ is the basal velocity, C is a sliding parameter, $\tau_b$ is the basal drag and m is the constant sliding exponent, set equal to 3.

For the Filchner-Ronne Ice Shelf, we extended the computational boundary some distance upstream from the grounding line, following the drainage basins of Zwally et al., (2012) and using the 2008-2009 ice front data from ALOS PALSAR and ENVISAT ASAR acquired during the International Polar Year, 2007-2009 (IPY) (Mouginot et al., 2017). Calving front positions for Pine Island Ice Shelf were extracted from Landsat 8 and Sentinel 1 A/B satellite imagery. Measurements of ice-flow velocities for Pine Island Ice Shelf were obtained from Joughin et al., 2021, for November 2019 and February 2020, while for Filchner-Ronne Ice Shelf observed velocities were derived from Landsat 8, Fahnestock et al., 2016, for the year 2014. Simulations were performed with an initial ice thickness, surface elevation and bedrock topography taken from the BedMachine Antarctica v2 dataset (Morlighem et al., 2020).

For each ice shelf, we inverted for the ice rate factor $A$. The inversion minimises a cost function ($J$) of the general form:

$$J = I + R;$$

1)

where I is the misfit term between modelled and observed velocities, given by:

$$I = \frac{1}{2Area}\int \left(\frac{u - u_{obs}}{u_{err}}\right)^2 dArea + \frac{1}{2Area}\int \left(\frac{v - v_{obs}}{v_{err}}\right)^2 dArea$$

2)

where $Area = \int dArea$ is the total area of the domain we are integrating on, $u$ and $v$ are the modelled horizontal $x$- and $y$- velocity components, respectively, $u_{obs}$ and $v_{obs}$ are the observed surface $x$- and $y$- velocity components, respectively, and the $u_{err}$ and $v_{err}$ are their relative observational errors, and $R$ is the regularisation term. The regularisation term follows the assumption that the prior probability density function can be described using a Matérn covariance function (Lindgren et al., 2011). Whenever performing an inversion in ice-flow models, the inverse solution (in this case, $A$) is affected by the set of constraints (regularisations parameters, $\gamma$) imposed in the regularisation term. The magnitude of $\gamma$ determines to what extent our inverse solution can deviate from the prior estimate imposed, such that a high $\gamma$ penalizes a solution that is substantially different from its prior. Here we imposed two regularisation parameters, $\gamma_s$ and $\gamma_a$, that penalized deviation of the parameters getting optimized from their prior estimates in terms of gradient and amplitude, respectively. The regularisation term takes the form:

$$R = \frac{1}{2Area}\left((p - \tilde{p})^T(\gamma_a^2 M)(p - \tilde{p})\right) + \frac{1}{2Area}\left((p - \tilde{p})^T \left(\gamma_s^2(D_{xx} + D_{yy})\right)(p - \tilde{p})\right)$$

3)

where $M$ is the mass matrix, $D_{xx} + D_{yy}$ is the stiffness matrix in x and y direction, respectively, $p = log_{10}(A)$, and $\tilde{p}$ is its relative prior estimate. To avoid the inverse solution either being shaped by the given prior or overfit observations, we adopted the L-curve method (Calvetti et al., 2002) which graphically visualises, in a log-log scale, the relationship between the norm of the regularised solution and the norm of the residual error. This technique consists of performing multiple simulations in which different regularisation magnitudes are tested and compared to their reciprocal norm residuals. The distribution of these

points follows an L-shape curve and the point where the horizontal and vertical branches converge is the L-corner. This point corresponds to the point of maximum curvature and represents a solution where the "perturbation" errors and the "regularisation" errors are well balanced. Since the regularisation equation that we solve in our inversions adopts two regularisation parameters, $\gamma_a$ and $\gamma_s$, we systematically have to assess both quantities for both regularisation parameters. Thus, we perform two sets of simulations: for each set, one of the two regularisation parameters is allowed to evolve by several

orders of magnitude, while the other is kept fixed, and the L-corner value is found. For our simulations, we found an ideal L-corner at $\gamma_a = 1$ and $\gamma_s = 25000$. Selecting a smaller value for $\gamma_s$ will not affect the calculated velocity distribution significantly, hence any further variation in $A$ resulting from selecting a smaller value is not supported by any corresponding variations in observed velocities. On the other hand, selecting very large values of $\gamma_s$ ($> 10^9$) will cause the solution of $A$ to be spatially uniform (we refer to Figure S5 in the Supplement material).

For the second step, we adopted two crevasse map products. For Pine Island Ice Shelf we used the 30 m resolution NormalisEd Radon transform Damage detection (NeRD) method, developed by Izeboud and Lhermitte, (2023), applied on a median composite sentinel S2 image for the austral summer of 2019-2020 (December, January, February), using red, green, and blue reflectance bands, and filter cloud cover < 20 %. Since the NeRD approach was validated just on the Amundsen Sea

Embayment, and given the limited changes observed in the Filchner Ronne Ice Shelf, we opted to use the CNN derived 125 m resolution crevasse map produced by Lai et al., (2020) for the Filchner-Ronne Ice Shelf, using a MODIS Mosaic from 2009 (Fig. 1).

The NeRD method applied on Pine Island provided a continuous damage map (D), which has values ranging from 0 for intact

ice, to 0.5 for fully damaged ice, with heavily damaged regions defined as areas with values greater than 0.1, containing heavily damaged ice, fully fractured rifts and ice mélange (Izeboud and Lhermitte, 2023). Hence, this continuous crevasse product can be converted into a binary grid, yielding two masks: one displaying all surface crevasses (D > 0) and the other exclusively highlighting heavily damaged crevasses (D > 0.1). This approach enabled us to assess two distinct scenarios for Pine Island Ice Shelf.

For the Filchner-Ronne Ice Shelf, on the other hand, we extracted the CNN product of Lai et al., (2020), which is already a binary data, either 0, for intact ice, or 1, for crevassed ice. Since the crevasse mask was obtained from MODIS MOA 2009, and the inverted ice rate factor fits observed velocities from 2014 (Fahnestock et al., 2016), we have tested the case for which

the 2009-crevasses were advected downstream for 5 years' worth distance, with constant velocity, for a better match with the 2014 ice rate field, but found no differences in the final results.

## 2.1 Classification techniques

In the third step of our methodology, we used a standard classification analysis (Tharwat, 2018) to test and evaluate the relationship between these two observational and modelled products. If there is a strong link between the inverted ice rate factor $A$ and the damage maps obtained from remote sensing methods, we would expect to be able to use the inverted $A$ field to classify areas of damage as identified through the remote sensing methods— predict crevasses where satellite maps detected crevasses (True Positive). Since we employed the SSA equations to invert for the solution of $A$, there are certain regions for which the SSA equations may not behave well / break down, i.e., in areas close to the grounding line, in regions where there's a drastic change in slope or topography, close to pinning points and ice rises, due to the presence of high vertical shear. To ensure reliable results from our ROC analysis we have excluded all areas of $A$ that were found within a 5 km radius of the grounding line.  To enable the comparison of these observational and modelled products in a common framework, we re-gridded one of the variables, to ensure consistent format and resolution. Since the binary crevasse mask is an equally spaced rectangular grid and the $A$ field is computed on a Delaunay triangulation mesh, we chose to interpolate the field $A$ onto the regular crevasse mask coordinates. Having achieved spatial consistency between the crevasse binary mask and the interpolated $A$ field, we proceeded with the assessment of their relationship via multiple classification analyses (Ferri et al., 2009; Yacouby and Axman, 2020).

Given that classification analyses generally involve comparing two categorical datasets, we need to transform the continuous $A$ field into a binary map, before comparing it with the binary crevasse map. However, whenever imposing an *A-threshold* above which ice is considered damaged, we considerably reduce the amount of information (in terms of spatial variability) present in $A$. To address this limitation, we employed Receiver Operating Characteristic (ROC) curve analyses (Hanley and McNeil, 1982; Bradley, 1997; Fawcett, 2006), which evaluate the classification performance — predictive capability — of each paired dataset ($A$ field vs crevasse map) across all potential classification *A-threshold*s. The strength of the ROC metric is its ability to evaluate the classifier's performance based on the True Positive Rate (TPR, "sensitivity", the proportion of true positives correctly classified as positive) and False Positive Rate (FPR, "specificity", the proportion of negatives that are incorrectly represented as positive) at any classification threshold. Here, the crevasse products obtained by remote sensing/machine learning techniques represent the true observations to be classified. The ROC analysis varies the classification threshold of the $A$ field, to generate a binary $A$ map for each threshold. Specifically, values of $A$ that exceed the designated classification threshold are categorised as damaged (1), while those that fall below this threshold are identified as intact ice (0). For each classification threshold, we compare the binary $A$ map with the binary crevasse map and produce a pair of TPR and FPR, corresponding to a dot in the ROC curve. The ideal perfect classifier corresponds to the right-angled line, vertical along the y axis, and parallel to the x axis, passing by the upper left corner of the plot, where TPR is 1 and FPR is 0 (red dot

in Fig. 2 and 3); A random classifier on the other hand corresponds to a straight diagonal line of the plot (dashed blue line in Fig. 2 and 3). The Area Under the Curve (AUC) is a metric that measures the overall performance of the classification analysis, and which describes the strength of the relationship between the two compared products, independently of the classification
threshold used (Rizk et al., 2019). An AUC value ranges from 0 to 1, where an AUC of 0.5 is equivalent to a random classifier, and an AUC of 1 is a perfect classifier. To further identify an ideal *A-threshold* which best compromises the FPR and TPR in this analysis, we calculate the OPTtimal operating PoiNT (OPT-PNT) on the ROC curve — further details on how this value is calculated in the ROC analysis can be found in the Supplementary material.

## 2.2 Adjusting unbalanced datasets

Since both crevasse maps (CNN and NeRD) have a greater sample size of "non-crevasse" nodes (99%) compared to the "crevassed" class, the dataset is, in this sense, highly unbalanced. Thus, careful evaluation and treatment is needed (Branco et al., 2015) when assessing the classifier performance in relation to the inverted *A* field (preliminary analysis in the Supplementary). When performing a ROC curve analysis on unbalanced datasets, results will be biased towards the majority class. To resolve this, we applied a standard approach in classification analysis of under-sampling the majority (non-crevasse)
class, to obtain an even distribution among classes (bias-corrected data). This is performed by randomly sub-sampling the majority class by the size of the minority class, thus forcing an equal sample size for both classes. Given the random nature of the sub-sampling procedure, we perform this step 2000 times, thus repeating the ROC curve analysis for each resampling iteration and recording all 2000 ROC curves and Area Under the Curve (AUC). We then calculate the OPT-PNT on each of the 2000 ROC curves, measuring the mean optimal *"A-threshold"*.

**3 Results**

Our inverse solution finds that the Filchner-Ronne Ice Shelf displays a consistently more stiff and resistant ice (low *A*) compared to Pine Island Ice Shelf, with softer ice (higher values) present just in the proximity of ice rises and ice rumples, at the ice shelf front, and in areas close to the grounding line (Fig. 1, panel a). In contrast, Pine Island Ice Shelf (Fig. 1, panel b) exhibits a visibly softer and more deformable ice (higher *A*), with values several orders of magnitudes larger than the *A* for
temperate ice along the ice shelf margins, specifically in the southern shear zone and in one region in the middle of the ice shelf. By visually comparing these *A* fields with their corresponding maps of remotely sensed crevasses, displayed on the right-hand side of each panel, we can already appreciate qualitatively the regions where these two fields overlap in contrast to regions where little or no agreement is present. A discrepancy is clearly visible between areas displaying an ice rate factor well below the reference *A* for temperate ice and the abundance of remotely sensed crevasses, particularly evident for Filchner-
Ronne Ice Shelf. The middle portion of Pine Island Ice Shelf, where the NeRD method detects a considerable number of crevasses, also exhibits a low ice rate factor (as shown in red boxes in Fig. S2b). Additionally, some regions where the ice rate factor exceeds that of temperate ice do not show remotely sensed crevasses, indicating that other factors, such as fabric, may be affecting changes in ice flow.


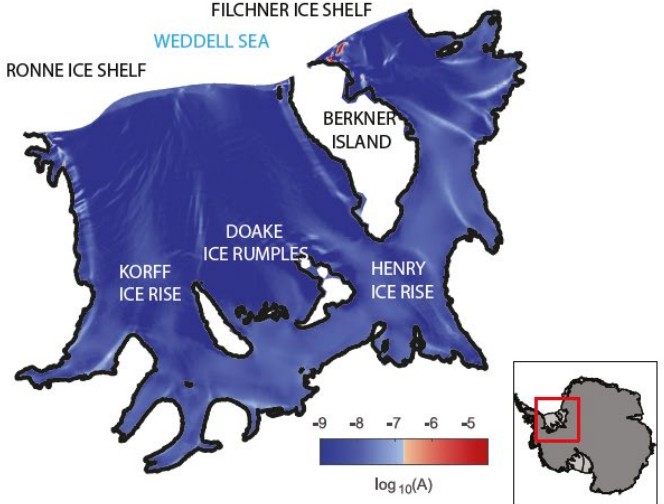

a) Ice rate factor A for Filchner-Ronne Ice Shelf

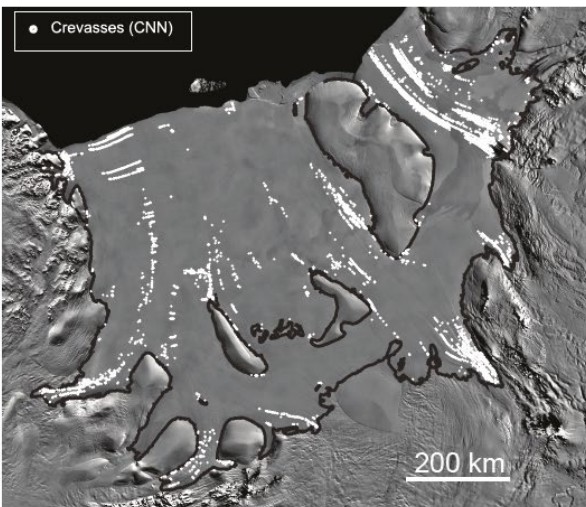

MODIS MOA 2009 with crevasses detected with CNN

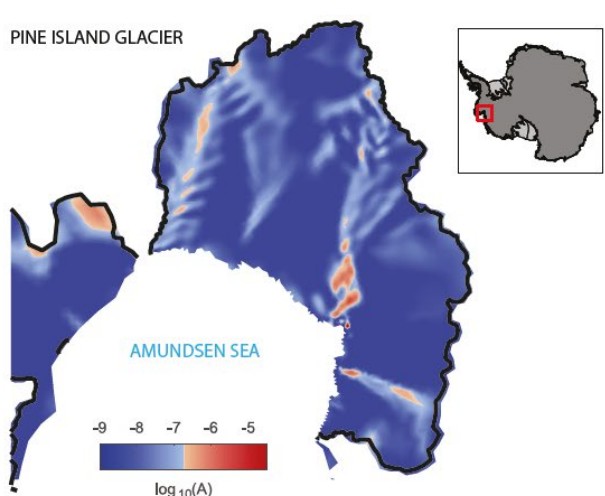

b) Ice rate factor A for Pine Island Ice Shelf

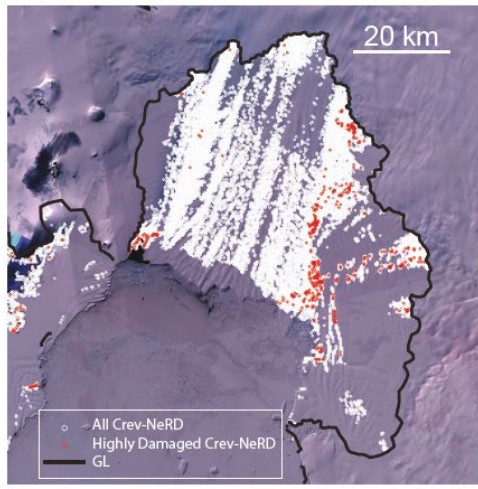

Sentinel 2 Austral Summer 2019-20 with crevasses detected with NeRD

**Figure 1: Summary of the inverted ice rate factor field for Filchner-Ronne (a) and Pine Island (b) ice shelves. To visually display the $A$ field and identify regions of damage, we have chosen to use a blue-white red colour scale, centred on the $A$ field for temperate ice obtained from experimental results (A = 1.67 x $10^{-7}$ KPa$^{-3}$a$^{-1}$, Spring and Morland, 1983, Cuffey and Paterson, 2010). This colour scale allows us to easily identify areas of damage by visually comparing the $A$ field to the reference value for temperate ice. Regions that have lower values of $A$ than the reference value indicate areas of less damage or intact ice (in blue), while regions with higher values indicate areas of damage (in red). On the right-hand side of each panel, a representation of the crevasses as detected with the CNN method (panel a, crevasses in white with MODIS mosaic 2009 underlayed), and all crevasses (in white) and heavily damaged**


**crevasses (in red) as mapped with the NeRD method (panel b, with Composite sentinel (S2) image of austral summer 2019-2020 underlayed) is displayed.**

### 3.1 Weak association between remotely sensed damage and inverted ice rate factor

The ROC curve results illustrated in Fig. 2 display the trade-off between the TPR and FPR of the binary classification models across different classification thresholds of $A$ for bias-corrected data for Filchner-Ronne Ice Shelf, for the 2009 crevasse map,

and for the case of the 2009 crevasses advected downstream for 5 years' worth of distance. Overall, we measure a mean-AUC for Filchner-Ronne Ice Shelf of 0.51 (with 95 % confidence intervals of 0.494 - 0.528) for the former and 0.52 (with 95 % confidence intervals of 0.501- 0.536) for the latter, demonstrating that the ROC's predictive performance is no better than a random classifier.

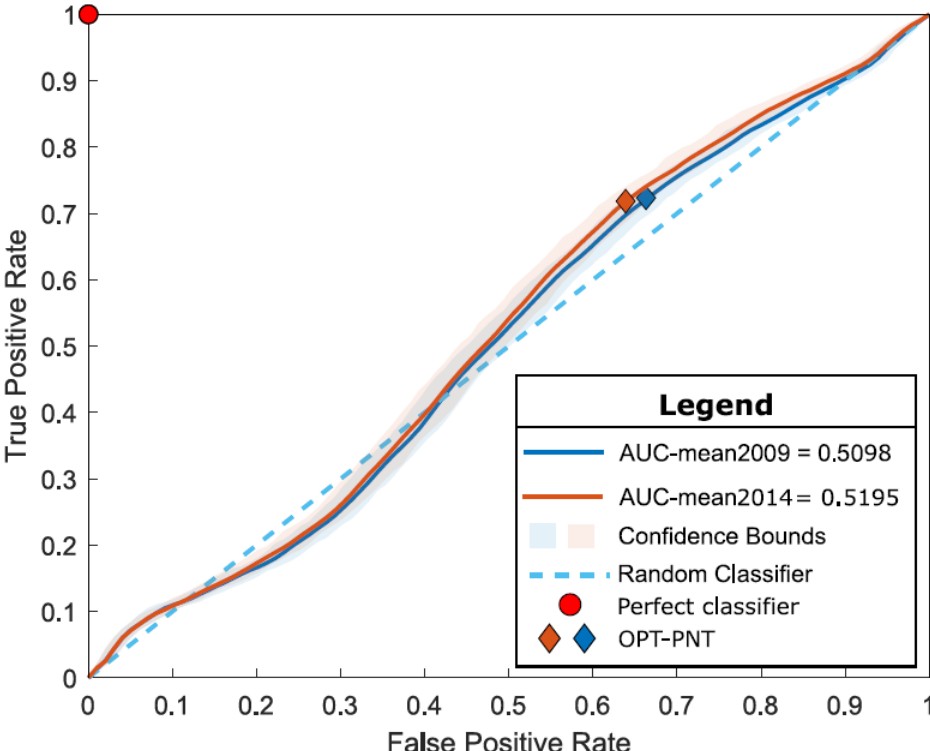


**Figure 2: ROC curve analysis for 2000 classification tests applied to Filchner-Ronne Ice Shelf, for all crevasses (2009, in blue) and for all crevasses advected downstream for 5 year-worth distance (in orange, to improve the match with the inverted ice rate factor when fitting the 2014 velocities). We compare the ice rate factor field, $A$, and the crevasse products obtained by remote sensing/machine learning techniques, which represent the true observations to be classified. The ideal perfect classifier corresponds**

**to the red dot; a random classifier corresponds to the diagonal line of the plot (dashed blue line). We find an AUC-mean of 0.51, that is slightly improved when considering the advected crevasses to an AUC-mean of 0.52. Both models suggest that the predictions are no better than a random classifier.**

The ROC curve analysis for Pine Island Ice Shelf, based on all surface crevasse features (Fig. 3a) and only heavily damaged

crevasses (Fig. 3b), for November 2019 (in blue) and February 2020 (in orange), showed slightly improved classification

performance compared to the Filchner Ronne ROC plot results. However, we found that the predictive performance is much

higher for the model based only on heavily crevassed regions, as opposed to that using all crevasses mapped by NeRD (Fig.

3, panel a and b). We found a mean-AUC of 0.55 for velocities of November 2019 — with 95 % confidence intervals of 0.542-

0.557 — and a mean-AUC of 0.55 for velocities of February 2020 — with 95 % confidence intervals of 0.539 -0.555 — for the

model based on all surface crevasses. The mean-AUC improves to 0.69 (November 2019) — with 95 % confidence intervals

of 0.649-0.724 — and 0.73, (February 2020) — with 95 % confidence intervals of 0.686- 0.765— for the model based only on

the heavily crevassed areas. These results show that surface features identified as heavily damaged crevasses are correlated to

a greater degree with the ice rate factor obtained through inversion methods. Nevertheless, the Area Under the Curve values

obtained still fall below the threshold for significance, as in classification analysis AUC values ranging from 0.7 to 0.8 are

typically interpreted as poor (Metz, 1978). We further found a larger uncertainty around the mean AUC for the heavily

damaged crevasse ROC analysis compared to that based on all surface crevasses. This reflects the former ROC analysis relying

on a smaller number of crevasses (412 heavily damaged crevasses) compared to the latter, where all crevasses are considered

(12466).

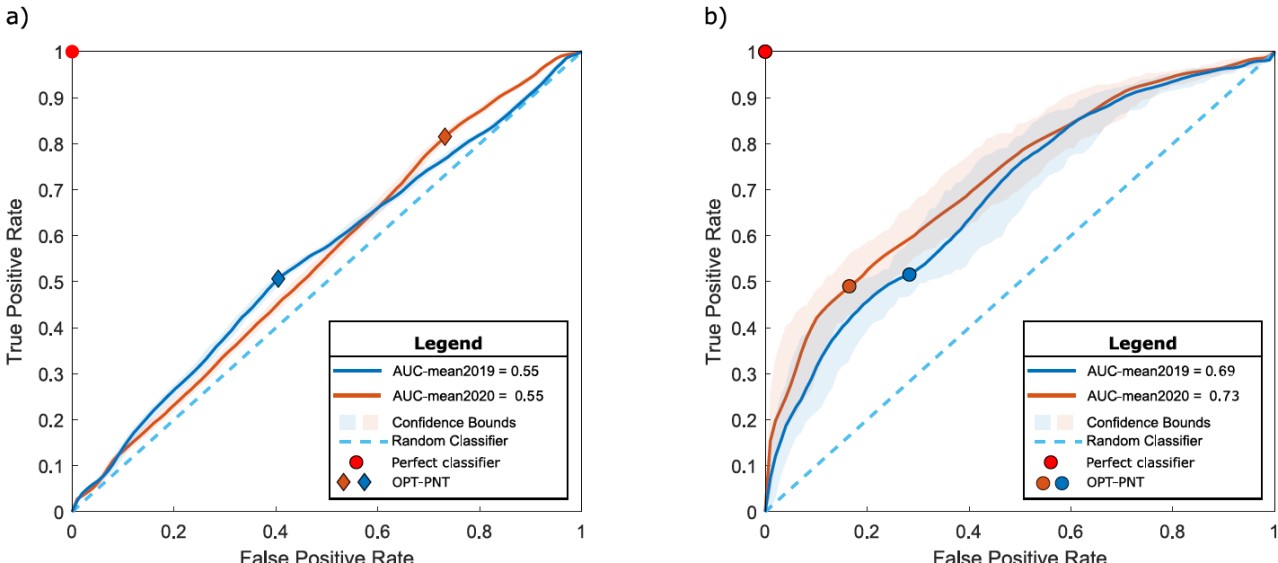

**Figure 3: ROC curve analysis for 2000 classification tests applied to Pine Island Ice Shelf, for all crevasses as mapped via NeRD (panel a) and for only heavily damaged crevasses (panel b). We find an AUC-mean of 0.55 and 0.73 when using an inverted ice rate factor obtained when fitting velocities for February 2020, respectively. This relationship is slightly reduced when considering the November 2019 velocities (an AUC-mean of 0.55 and 0.69, respectively). Greater uncertainty is estimated for the case of heavily damaged crevasses (0.67- 0.77). These results suggest that the model's predictions are improved, to some extent, when considering**
**just heavily damaged crevasses.**

## 3.2 Results are robust to changes in the regularisation parameter ($\gamma_s$)

The ROC plot's classification analysis hinges on the accuracy of the currently available crevasse maps derived by satellite images and on the ice rate factor field $A$, which cannot be directly observed, and must be derived through inverse methods. The reliability of the inverted $A$ field can be influenced by two key factors. Firstly, errors can arise from the observed velocities

during the inversion process when minimising the discrepancies between them and the modelled velocities. Secondly, the spatial distribution of the inverted $A$ field may be affected by the choice of the regularisation parameters ($\gamma_s$) used to refine the gradient of $A$ based on prior information, during the inversion. In this aspect, while we assumed the velocity data provided by the remote sensing community to be reliable for the first factor, we tested the second by evaluating the sensitivity of our classification results to the choice of $\gamma_s$. The sensitivity analysis was assessed in the context of Pine Island Ice Shelf, for both

November 2019 (in Supplementary) and February 2020 velocities, and the classification results, inclusive of all heavily damaged crevasses, are presented in Fig. 4 and Fig. S3. With $\gamma_a$ fixed at a constant value, we systematically explored a spectrum of $\gamma_s$ values in accordance with the values used by the L-curve method (Fig. 4a). Notably, as $\gamma_s$ is progressively increased (Fig. 4a and b) to values $> 10^6$, the model's predictive capability (AUC-mean) is reduced (closer to the random classifier dashed line). This outcome is expected, since an increase in $\gamma_s$ enforces a more pronounced smoothing effect on the inverted $A$ field,

thus reducing the correlation between inverted and remotely sensed damage. However, this is not the case for the range of acceptable values of the regularisation parameter — $\gamma_s$ within the L-corner range— within which the ROC plot results remain invariant.

### 3.3 Consistent poor classification ability even for optimal *A-threshold*

While these results provide insights into the relationship between inferred $A$ and the observed crevasse map for any possible threshold value in $A$, it may be beneficial to identify an optimal *A-threshold*, specifically for the case of heavily damaged crevasses, where this relationship is strongest. When performing this investigation on Pine Island Ice Shelf, for values of $\gamma_s$ in

the L-corner range, we found a median optimal *A-threshold* for heavily damaged crevasses of 2.06 $*10^{-8}$ $a^{-1}kPa^{-3}$ with a minimum and maximum range [1.36$*10^{-8}$ 2.39$*10^{-8}$] $a^{-1}kPa^{-3}$ (Fig. 4c). Additionally, when adopting this threshold, the classifier accurately identifies 50 % of the crevasses, while also incorrectly flagging almost 20 % of no-crevasse regions as crevasses (Fig. 5). While it is plausible that values above this *A-threshold* may indicate damage for this ice shelf, the probability of a match between inverted and remotely-detected damage remains limited to 50 %.


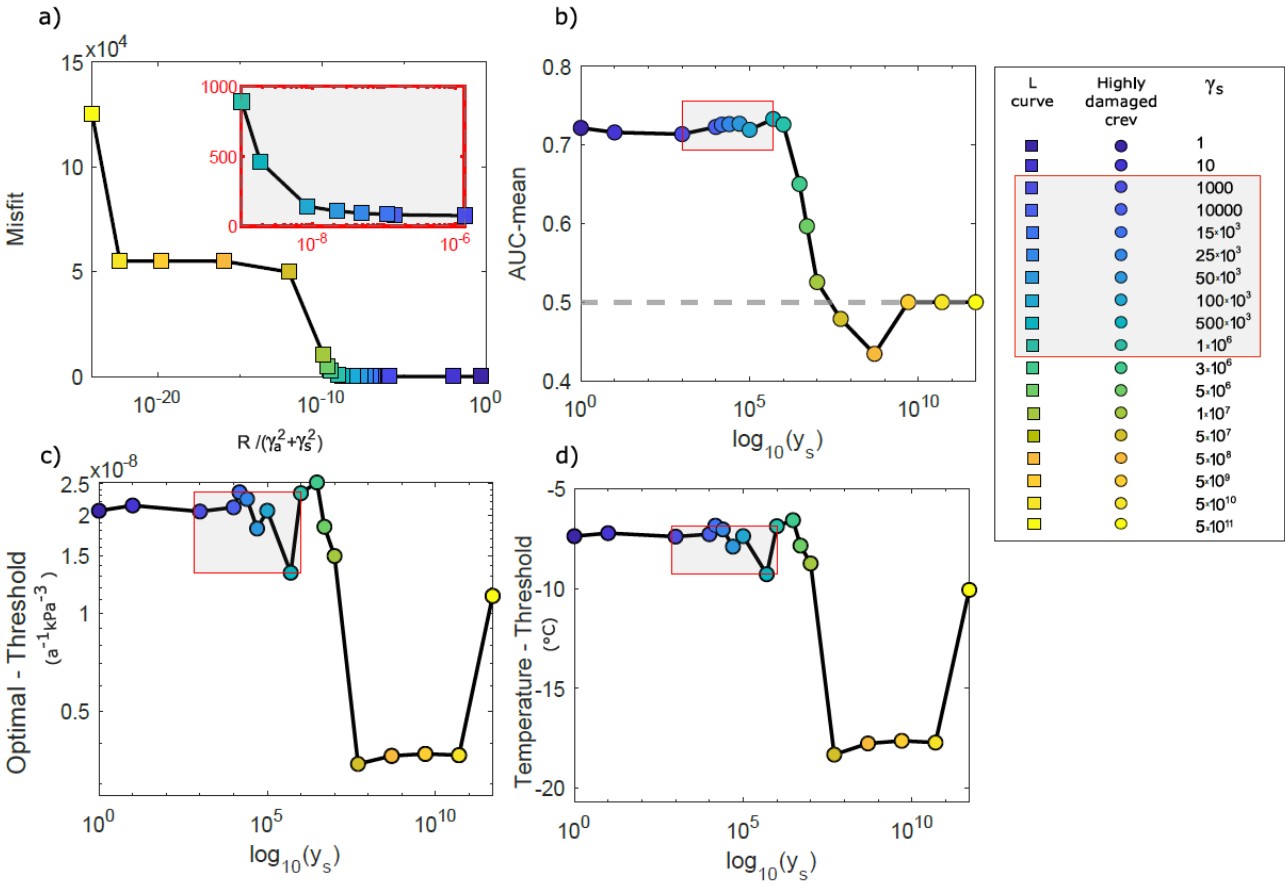

**Figure 4: a)** L-curve analysis for the Pine Island Ice Shelf, with a zoomed-in inset highlighting the L-corner range. **b)** AUC-mean values for heavily damaged crevasses across varying regularisation parameters ($\gamma_s$). The AUC-mean is reduced for $\gamma_s$ values exceeding $10^6$, as a smoother ice rate factor is applied. **c)** and **d)** Optimal *A-threshold* determined from the ROC plots and corresponding temperature derived from Glen's Flow Law for heavily damaged crevasses, as a function of $\gamma_s$. Within each panel, a red-outlined box filled with grey shading highlights the acceptable regularisation values for $\gamma_s$, for heavily damaged crevasses. The AUC-mean values, calculated from ROC analyses that adopted simulations with regularisation values of $\gamma_s$ spanning the range [$10^2$ to $10^6$], exhibit magnitudes tightly clustered around a common value, ~ 0.73. Nevertheless, as we progressively introduce larger $\gamma_s$ values (exceeding $10^6$), there is a discernible and consistent reduction in the magnitude of the AUC-mean. Variations in $\gamma_s$ values in the L-corner range do not significantly impact the robustness of our results.

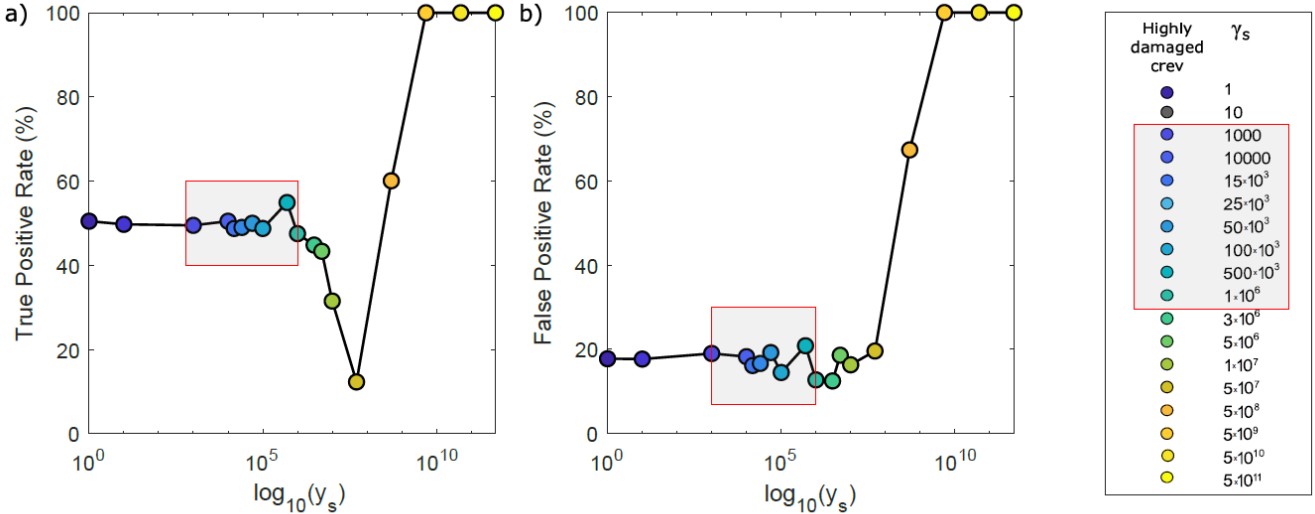

**Figure 5: True Positive Rate (a) and False Positive Rate (b) for Pine Island Ice Shelf across varying regularisation values ($\gamma_s$), for heavily damaged crevasses for simulations that fitted velocities for February 2020, for the best optimal A-threshold. Within each panel, a red-outlined box filled with grey shading highlights the acceptable values for $\gamma_s$—within the L-corner range.**


## 4 Discussion

An abundance of studies has been dedicated over the past decade to detect crevasses across Antarctica using remote sensing and machine learning techniques, but it remains uncertain whether modellers can incorporate these products to inform the rheology state of ice and constrain future sea level rise projections. In this study, we have analysed whether the inferred

distribution of the ice rate factor $A$ of an ice flow model, which by construction is consistent with the observed surface velocities, agrees with the damage maps derived from surface morphological features. Overall, we find a poor relationship between these observational and inverted products. However, when we specifically target regions where remotely derived damage maps display heavy crevassing, such as the shear margins of Pine Island Ice Shelf, the two products are somewhat correlated. Yet, the calculation of an optimal *A-threshold* which best compromises the ROC analysis, has shown that even in

this setting, the association between inverted and observational products is limited.

Several studies (MacGregor et al., 2012; Alley et al., 2019; Lhermitte et al., 2020) have documented the increase of areas affected by damage on Pine Island Ice Shelf over the time period from 2017 to 2020, with a sustained series of calving events and high localized fracturing occurring at the ice shelf margins (Lhermitte et al., 2020; Joughin et al., 2021). The classification

analysis performed in this study showed an improved yet modest relationship when specifically targeting heavily crevassed regions, in Pine Island Ice Shelf's margins (AUC = 0.73 for the 2020 ice velocities). The relationship is poor, however, when targeting all features identified as crevasses. Qualitatively, there are multiple regions where this discrepancy is clearly visible:

for instance, those crevasse-like features located in the centre of Pine Island Ice Shelf, where a stiffer and less deformable ice is inferred (red boxes in Fig. S2b); Equally, in some regions of the Filchner-Ronne Ice Shelf the ice rate factor $A$ is lower than

that of temperate ice, yet there is a high abundance of crevasses detected through remote sensing images. Other regions display the opposite relationship.

The classification analysis further provided us with an optimal *A-threshold*, $2.06*10^{-8}$ $a^{-1}kPa^{-3}$, where the relationship between inverted and observational products is strongest. If we translate this *A-threshold* value into ice temperature (°C) using Glen's

flow law for a stress exponent n = 3, and following the approach of Spring and Morland, 1983, we find that it corresponds to a mean temperature of approximately -7.4°C, with a minimum and maximum range of [-9.1 -6.7] °C, a temperature substantially lower than that of temperate ice (Fig. 4d). Since the model employs a 2D depth integrated formulation, resulting in a depth integrated ice rate factor and temperature, this value will not capture the natural variation of ice shelf temperature with depth. Thus, it becomes challenging to interpret this optimal temperature, and extrapolate this critical *A-threshold* for

future modelling purposes.

The observed discrepancy between modelled damaged and remotely sensed crevasse maps is not to be attributed to deficiencies in the physics incorporated within the model, such as the use of a depth-integrated approximation or the omission of the elastic component of deformation. Our methodology, which leverages SSA equations and inversion techniques, has proven effective

in identifying rift formation and pinpointing their locations (De Rydt et al., 2019), yielding an inverted solution that not only accurately located areas of weakening but also faithfully replicated independent analyses of ice rheology (King et al., 2018). Our study abstained from investigating elastic effects, firstly as the predominant behaviour of ice resembles that of a viscous fluid, and instances of elastic behaviour arise in limited settings; secondly, as elastic effects can be safely disregarded at stresses and strain rates typical of ice shelf flow — given that loading periods and strain rates in ice shelves are approximately five

orders of magnitude too small for elastic effects to be significant (Gudmundsson, 2007). The limited agreement between the two damage products suggests that the majority of surface crevasses identified through satellite imagery are shallow features that do not exert a discernible impact on the depth-integrated ice viscosity and ice flow.

Since remotely sensed crevasses do not have a strong association with the inferred ice rate factor $A$, it is worth questioning

whether the extensive effort placed into tracking crevasses across the whole of Antarctica is of direct relevance to modelling efforts regarding future sea-level rise projections. The inverted ice rate factor $A$ that ice sheet modellers obtain by fitting horizontal surface velocities provides a quite detailed and comprehensive understanding of the depth-integrated properties of the ice for that moment in time (Albrecht and Levermann, 2014; Borstad et al., 2012, 2013; Khazendar et al., 2007). Horizontal stresses cannot be effectively transferred across highly fractured regions, due to the loss of mechanical integrity and load-

bearing capacity (Borstad et al., 2012, 2013). So, if a crevasse is active and affecting ice flow, the strain rates and velocities in that region will change, which will be detected by an ice-sheet model as a local increase in the ice rate factor parameter $A$. On

the other hand, crevasses that were previously active and which later became passive or were later advected in regions of compression, are detected and expressed by lower values of $A$. Additionally, regions that display inferred high values of $A$ may not necessarily match areas of active rifts, thus providing insights into the structural integrity of the ice at that location

and highlighting the presence of other dynamical processes currently occurring.

Current ice flow models do not account for an evolving ice rate factor in transient simulations, thus the accuracy of their projections remains restricted. Recent advances adopting a novel physic informed machine learning framework have tested a complementary approach to calibrate uncertainties in the ice rate factor and sea level forecasts, by inferring a posterior

distribution of the ice rate factor (Riel and Minchew, 2023), providing some insights on better and continuously updated calibrations of ice flow parameters. To further reduce these errors in the ice rate factor field when adopting more conventional inversion procedures, it remains crucial for future efforts to prioritize the monitoring of ice shelves' velocity, geometry, and calving front positions, which are essential parameters for accurate estimates of sea level forecasts.

## 5 Conclusions

We have investigated the relationship between inverted values of rheological parameters used in ice-flow models and independent estimates of ice damage extracted from satellite imagery. In particular, we have critically evaluated the assertion that estimates of ice damage can be used to inform ice-flow modelling studies by providing constraints on the rheological parameter values ($A$) used in those models. Our key measure is the ability to correctly predict the existence of ice damage in each area from our inverted $A$ values, and correctly predict no damage where none is found. To do this we must select a

threshold value for $A$ above which the ice is considered damaged. We find that for *any* threshold value, the performance of this predictor is like that of a random classifier. While areas for which our inverted $A$ fields are above a given threshold value (suggesting ice damage) often coincide with areas where analyses of satellite imagery indicate structurally compromised ice (i.e., true positives), in many other locations where our inverted $A$ fields are above that same threshold value, the satellite products suggest undamaged ice (i.e., false positives). In essence, we do not find a clear relationship between damage inferred

by an ice sheet model and damage identified via remote sensing.

Our estimates of the spatial distribution of the rheological parameter $A$ are based on model inversions of measured velocities, and therefore reflect spatial variation in the properties of ice that have a measurable impact on ice flow. Any feature identified as damage through other methods that does not correspond to areas that a model identifies as softer ice does not, by extension,

have a measurable impact on ice flow. Our results show that some of the variability in estimated ice damage, and crevasse densities, derived from analysis of surface morphological features, may not be impacting the flow of the ice. This would, for example, be the case if some of the observed crevasses are passive features. Conversely, we also find that our model inversion implies areas affected by ice damage (e.g., areas where the ice rate factor $A$ is for example higher than that of temperate ice), where no surface damage or crevasses have been identified.


This lack of a match between ice damage and crevasse density and inferred variation in ice rheological parameters warrant further studies. It seems intuitively plausible that surface crevasses on ice shelves may penetrate to sufficient depths to hit the water line and sound theoretical arguments support this expectation, especially in the presence of surface water input (Lai et al., 2020). Indeed, we do find areas, such as within the shear margins of Pine Island Ice Shelf, where crevasse density and

inferred $A$ values both indicate damaged ice. This suggests that further work might be able to elucidate the reasons for the lack of correspondence between these independent estimates of damage.

## 6 Code availability

The source code of the Úa ice-flow model, sufficient to run the experiments in this project, is available at

https://doi.org/10.5281/zenodo.3706624.

## 7 Data availability

The crevasse mask product of Lai et al., (2020), is available at: https://doi.org/10.15784/601335. The code of the NeRD method developed by Izeboud and Lhermitte, (2023), is available on github,

https://github.com/mizeboud/NormalisedRadonTransform. Input datasets used in our experiments were the Antarctic drainage basins (Zwally et al., 2012), geometry from Bedmachine Antarctica v2 data set (Morlighem et al., 2020), 2008–2009 ice-front data from ALOS PALSAR and ENVISAT ASAR (Mouginot et al., 2017), Landsat 8 observed velocities for the Filchner Ronne Ice shelf (Fahnestock et al., 2016), and observed velocities for Pine Island Ice Shelf were downloaded from Joughin et al., (2021) on https://digital.lib.washington.edu/researchworks/handle/1773/46687.


## 8 Author Contributions

CG, SR and HG designed the study, CG and SS ran the model simulations with Úa. CG performed the classification analysis and produced the figures. All authors contributed ideas to interpretation of the results and to the writing of the article.


## 9 Declaration of Competing Interest

The authors declare that they have no real or perceived competing financial interests.

## 10 Funding

This project has received funding from the European Union's Horizon 2020 research and innovation programme under grant agreement no. 820575. This work was also funded through the PROPHET project, a component of the International Thwaites Glacier Collaboration (ITGC) by the Natural Environment Research Council under grant agreement NE/S006745/1. ITGC Contribution No. (number to be added ahead of final publication).

This manuscript is respectfully dedicated to the memory of Chris Borstad, a distinguished pioneer in glaciology and damage mechanics, who passed away in November 2023. His significant contributions have greatly influenced our understanding of this field and serve as a source of inspiration that continues to shape and guide the trajectory of our work.

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
