# Peer review of "Weak relationship between remotely detected crevasses and inferred ice rheological parameters on Antarctic ice shelves"

_EGUsphere, 2023_

## Referee Comment (RC3)

[referee-annotated manuscript omitted]

---

## Author Comment (AC2)

Response to Reviewer1:
*Weak relationship between remotely detected crevasses and inferred ice rheological parameters on Antarctic ice shelves; C., Gerli, S., Rosier, H., Gudmundsson, S., Sun*

**Anonymous Referee #1, 31st Dec 2023**

**Improve the image quality and readability of some figures.**

- Figures 1 and S2 can be difficult to read and would benefit from higher resolution.

Response: The figures in the Word document are at lower resolutions but a higher resolution vector format PDF version will be produced for each figure and will be submitted with the resubmitted article.

- Figures 1 and S2 have difficult to read legends partially due to resolution and partially due to the low contrast between text color and background color.

Response: Legend text will be made bigger and easier to read

- Figure S1: I recommend changing the top left box in each matrix to a lighter background color with better contrast. It can be difficult to read the numbers on a printed copy.

Response: Light blue color for top left box of matrix will be applied

**Include some additional technical details to better understand the study's methods.**

- Line 160: you interpolate the field "A" onto the regular crevasse mask coordinates. Could you briefly name or explain the interpolation method?

Response: The interpolation method was performed using a linear interpolation which is appropriate for the linear elements used in the model.

- Are results robust to changes in the other regularization parameter? The paper focused on sensitivity in one parameter, but not the other.

Response: We will perform a second set of simulations for a $\gamma_a = 0$ instead of $\gamma_a = 1$, which allows A to change in terms of its magnitude, fitting the velocities of 2019. We will add an additional figure in the Supplementary material displaying the sensitivity analysis to $\gamma_a$.

- Is there a numerical threshold or method for determining the "consistent agreement" mentioned at the end of the Figure 4 caption?

Response: We will change "consistent agreement" in the caption and clarify with further text.

In Figure S3(b) for the highly damaged case, the ROC curves become close to random classifiers at high values of the regularization parameter. However, the opposite is true for the "all crevasse" case in Figure S3(a). What is the reason for that behavior?

Response: By adopting larger $\gamma_s$ values in the inversion problem, the solution of A becomes more and more uniform throughout the domain (we will add a figure in the Supplement Material regarding this). For the cases of total uniformity, until the A threshold is not surpassed, all crevasses remain undetected. Once the threshold is surpassed, the model accurately identifies all crevasses, but at the cost of misclassifying all the noncrevassed areas as crevassed. This, therefore, provides a ROC curve that is a straight diagonal line from the bottom-left (0-0) to the top-right (1-1), indicating that the model's performance is as good as random, and providing a 100% TPR and 100% FPR. For the cases in the ROC analysis where we see an S-shaped curve, i.e., $\gamma_s = 10^7$ or $5*10^8$ (We will add a figure in the Supplement), the solution of A has not yet reached a total uniformity changing just slightly throughout the domain. Whenever performing the ROC analysis, the predictions and the misses are not 100% but are still split, thus the estimated curve takes the form of an s-shaped one. An s-shaped ROC curve represents a biphasic behaviour, reflecting different levels of classification at different thresholds, which does not necessarily mean a better classifier, but rather a two-phase discriminative ability as a function of the threshold. However, as uniformity becomes more pronounced (and A becomes spatially uniform), the ROC curve transitions towards resembling that of a diagonal line, characteristic of a random classifier. This transition underscores the diminishing ability of the classifier as it struggles to differentiate between classes effectively (crevasse and no-crevasse). We will make sure this is clear in the Supplementary.

**Provide some additional context/clarity for some background information and conclusions.**

- Lines 35-38: Can you provide citations for the impact of surface crevasses in different situations?

Response: We will provide some citations.

- I was confused by the "(see details below)" note in line 48 and the "(described further below)" in line 56. Can specific sections or subsections be referenced to make that information more clear?

Response: We will add it in the text.

- I find it interesting that for the analysis on Pine Island Ice Shelf including only heavily damaged crevasses (Figure 3b), the relationship no longer behaves like a random classifier. I would be interested to know what attributes "heavily damaged crevasses" have that other crevasses lack in the input data. I think that some brief discussion of the input data's classification method could provide valuable context to the "heavily damaged" result.

Response: For the heavily damage crevasses map, we adopt the methodology of Izeboud et al., 2023, which classifies areas of surface structural damage on ice shelves using multi-source satellite imagery through a feature contrast approach. This methodology provides a continuous damage map (D), which has values ranging from 0 for intact ice, to 0.5 for fully damaged ice, with heavily damaged regions defined as areas with values greater than 0.1. We will add it to the text in the manuscript.

Is there a different well-motivated choice in sliding law or sliding law exponent (e.g. regularized Coulomb friction) that would affect the results of this study? How dependent is the spatial variation of "A" dependent on the sliding law for this study?

Response: The inversions were performed adopting a Weertman sliding law for the grounded ice as our model domain extends beyond the floating shelf. We will not test the application of other sliding laws since the aim was to perform inversions to find the solution

of A on the floating ice, which relies on the strain rates and velocity. There is therefore no necessity to evaluate the impact of sliding laws for this analysis.

- Lines 365-366: You state "We find that for any threshold value, the performance of this predictor is like that of a random classifier". That seems inconsistent with the "high damage" study in the manuscript. I don't think that the "high damage" study weakens the paper's overall conclusions, but is an interesting result that perhaps merits some additional discussion.

Response: Indeed, these results show that surface features identified as heavily damaged crevasses are correlated to a greater degree with the ice rate factor obtained through inversion methods. However, the AUC values are still not satisfactory enough to be considered significant since, in classification analysis, any AUC value between 0.7 and 0.8 is generally interpreted as poor (Metz et al., 1978). If we still were to assume this was satisfactory, and we adopted the A-threshold given by the ROC curve for this case, the classifier accurately identifies 50 % of the crevasses but also incorrectly flags almost 20 % of no-crevasse regions as crevasses (Fig. 5). The probability of a match between inverted and remotely-detected damage remains limited even in this setting, so we believe that these results do not show any clear or robust relationship between damage inferred by an ice sheet model and damage identified via remote sensing. We will make it clear in the text.

Lines 382-384: you mention the possibility of surface crevasses penetrating to the water line. Can you give additional context to the correlation between surface crevasses reaching the water line and inferred variation in ice rheology?

Response: The theoretical findings by Lai et al. in 2020 highlighted the possibility of surface crevasses extending down to the water line. Here, we have not and will not investigate this; further information regarding the crevasses reaching the water line can be found at Lai et al., 2020. The inverted ice rate factor A obtained in our work by fitting observed velocities is a depth-integrated solution. Our results show ice weakening along the ice shelf margins, specifically of Pine Island Ice Shelf, where crevasses and damaged ice are present and expected.

- I quite like Figure S6 as a visualization for the classification. Can a similar figure for the Filchner-Ronne Ice Shelf be included as well?

Response: We will add an additional figure for the Filchner Ronne Ice Shelf in the supplementary as requested.

**Technical Corrections:**

- The acronym OPTtimal operating PoinT (OPTPT) defined in the text (lines 184 and 197) differs from the acronym used in Figures 2 and 3 (OPT-PNT), assuming the two have the same meaning like I believe they do. We will correct it.

- Table T2: Column 2, Row 3 has an extra "tab" space that should be deleted. We will correct it.

- Notation for the equation in section 1.3 of the supplementary information should be made consistent. Most of the terms use forward slashes but "Cost(P|N)" uses a vertical bar in the text. We will correct it.

---

## Author Response (AR1)

Response to reviewers:
*Weak relationship between remotely detected crevasses and inferred ice rheological parameters on Antarctic ice shelves; C., Gerli, S., Rosier, H., Gudmundsson, S., Sun*

We thank Adrien Gilbert and two anonymous reviewers for their helpful and insightful comments. A response to each reviewer follows in blue text, with bold text to outline what was added to the manuscript. Line numbers refers to the tracked changes file.

**Anonymous Referee #1, 31st Dec 2023**

**Improve the image quality and readability of some figures.**

- Figures 1 and S2 can be difficult to read and would benefit from higher resolution.

Response: The figures in the Word document are at lower resolutions but a higher resolution vector format PDF version can be provided with the resubmitted article.

- Figures 1 and S2 have difficult to read legends partially due to resolution and partially due to the low contrast between text color and background color.

Response: Legend text was made bigger and easier to read

- Figure S1: I recommend changing the top left box in each matrix to a lighter background color with better contrast. It can be difficult to read the numbers on a printed copy.

Response: Light blue color for top left box of matrix applied

**Include some additional technical details to better understand the study's methods.**

- Line 160: you interpolate the field "A" onto the regular crevasse mask coordinates. Could you briefly name or explain the interpolation method?

Response: The interpolation method was performed using a linear interpolation which is appropriate for the linear elements used in the model.

- Are results robust to changes in the other regularization parameter? The paper focused on sensitivity in one parameter, but not the other.

Response: Given that we do not want to constrain the final solution of A in terms of its magnitude, we have run simulations for a $\gamma_a$ = 1. We conducted a second simulation to evaluate the effect of using $\gamma_a$ = 0 instead of $\gamma_a$ = 1, fitting the velocities of 2019. Overall, our findings suggest that the discrepancies in our results are negligible. An additional figure (Figure S10) was added in the Supplementary material displaying the sensitivity analysis.

- Is there a numerical threshold or method for determining the "consistent agreement" mentioned at the end of the Figure 4 caption?

Response: We changed "consistent agreement" in the caption (line 320-324): "**the AUC-mean values, calculated from ROC analyses that adopted simulations with regularisation values of $\gamma_s$ spanning the range [$10^2$ to $10^6$], exhibit magnitudes tightly clustered around a common value, approximately 0.73. Nevertheless, as we progressively introduce larger $\gamma_s$ values (exceeding $10^6$), there is a discernible and**

**consistent reduction in the magnitude of the AUC-mean. Variations in γ$_s$ values in the L-corner range do not significantly impact the robustness of our results.** "

In Figure S3(b) for the highly damaged case, the ROC curves become close to random classifiers at high values of the regularization parameter. However, the opposite is true for the "all crevasse" case in Figure S3(a). What is the reason for that behavior?

Response: By adopting larger γ$_s$ values in the inversion problem, the solution of A becomes more and more uniform throughout the domain (please refer to the additional figure S5 in the Supplement Material). For the cases of total uniformity, until the A threshold is not surpassed, all crevasses remain undetected. Once the threshold is surpassed, the model accurately identifies all crevasses, but at the cost of misclassifying all the non-crevassed areas as crevassed. This, therefore, provides a ROC curve that is a straight diagonal line from the bottom-left (0-0) to the top-right (1-1), indicating that the model's performance is as good as random, and providing a 100% TPR and 100% FPR (Figure 5 and Figure S6). For the cases in the ROC analysis where we see an S-shaped curve, i.e., γ$_s$ = $10^7$ (Figure S4 a) in Supplement), the solution of A has not yet reached a total uniformity (Figure S4 b), changing just slightly throughout the domain. Whenever performing the ROC analysis, the predictions and the misses are not 100% (Figure S6) but are still split, thus the estimated curve takes the form of an s-shaped one. An s-shaped ROC curve represents a biphasic behaviour, reflecting different levels of classification at different thresholds, which does not necessarily mean a better classifier, but rather a two-phase discriminative ability as a function of the threshold. However, as uniformity becomes more pronounced (and A becomes spatially uniform), the ROC curve transitions towards resembling that of a diagonal line, characteristic of a random classifier. This transition underscores the diminishing ability of the classifier as it struggles to differentiate between classes effectively (crevasse and no-crevasse). We have added text in the Supplementary.

**Provide some additional context/clarity for some background information and conclusions.**

- Lines 35-38: Can you provide citations for the impact of surface crevasses in different situations?

We have provided some citations (now line 37-38). Citations added: **Weertman, 1983; van Der Veen, 1998; Larour et al., 2004; van Der Veen, 2007, Khazendar et al., 2009, Colgan et al., 2016.**

Response:

- I was confused by the "(see details below)" note in line 48 and the "(described further below)" in line 56. Can specific sections or subsections be referenced to make that information more clear?

Response: We opted not to further divide the methodology into subsections; instead, we refer in the text to Section 2 - the method (included in the text) where Glen's flow law and regularisation are defined.

- I find it interesting that for the analysis on Pine Island Ice Shelf including only heavily damaged crevasses (Figure 3b), the relationship no longer behaves like a random classifier. I would be interested to know what attributes "heavily damaged

crevasses" have that other crevasses lack in the input data. I think that some brief discussion of the input data's classification method could provide valuable context to the "heavily damaged" result.

Response: For the heavily damage crevasses map, we adopt the methodology of Izeboud et al., 2023, which classifies areas of surface structural damage on ice shelves using multi-source satellite imagery through a feature contrast approach. This methodology provides a continuous damage map (D), which has values ranging from 0 for intact ice, to 0.5 for fully damaged ice, with heavily damaged regions defined as areas with values greater than 0.1. We have added to the text, at line 155-6: **containing heavily damaged ice, fully fractured rifts and ice mélange (Izeboud and Lhermitte, 2023).**

- Is there a different well-motivated choice in sliding law or sliding law exponent (e.g. regularized Coulomb friction) that would affect the results of this study? How dependent is the spatial variation of "A" dependent on the sliding law for this study?

Response: The inversions were performed adopting a Weertman sliding law for the grounded ice as our model domain extends beyond the floating shelf. We did not test the application of other sliding laws since the aim was to perform inversions to find the solution of A on the floating ice, which relies on the strain rates and velocity. There is therefore no necessity to evaluate the impact of sliding laws for this analysis.

- Lines 365-366: You state "We find that for any threshold value, the performance of this predictor is like that of a random classifier". That seems inconsistent with the "high damage" study in the manuscript. I don't think that the "high damage" study weakens the paper's overall conclusions, but is an interesting result that perhaps merits some additional discussion.

Response: Indeed, these results show that surface features identified as heavily damaged crevasses are correlated to a greater degree with the ice rate factor obtained through inversion methods. However, the AUC values are still not satisfactory enough to be considered significant since, in classification analysis, any AUC value between 0.7 and 0.8 is generally interpreted as poor (Metz et al., 1978). If we still were to assume this was satisfactory, and we adopted the A-threshold given by the ROC curve for this case, the classifier accurately identifies 50 % of the crevasses but also incorrectly flags almost 20 % of no-crevasse regions as crevasses (Fig. 5). The probability of a match between inverted and remotely-detected damage remains limited even in this setting, so we believe that these results do not show any clear or robust relationship between damage inferred by an ice sheet model and damage identified via remote sensing. We have made it clear in the text, line 269-71. **"Nevertheless, the Area Under the Curve values obtained still fall below the threshold for significance, as in classification analysis, AUC values ranging from 0.7 to 0.8 are typically interpreted as poor (Metz, 1978)."**

- Lines 382-384: you mention the possibility of surface crevasses penetrating to the water line. Can you give additional context to the correlation between surface crevasses reaching the water line and inferred variation in ice rheology?

Response: The theoretical findings by Lai et al. in 2020 highlighted the possibility of surface crevasses extending down to the water line. Here, we do not investigate this; further information regarding the crevasses reaching the water line can be found at Lai et al., 2020. The inverted ice rate factor A obtained in our work by fitting observed velocities is a depth-integrated solution. Our results show ice weakening along the ice shelf margins,

specifically of Pine Island Ice Shelf, where crevasses and damaged ice are present and expected.

- I quite like Figure S6 as a visualization for the classification. Can a similar figure for the Filchner-Ronne Ice Shelf be included as well?

Response: An additional figure for the Filchner Ronne Ice Shelf was included in the supplementary as requested (Figure S8). We found a small mistake in the code for the Pine Island figure (former Figure S6 a, now Figure S7 a) as the threshold used for all crevasses, was the threshold obtained for the heavily damaged crevasses. We have corrected it and replotted the map with the correct threshold value. Apologies and many thanks for drawing our attention again here.

**Technical Corrections:**

- The acronym OPTtimal operating PoinT (OPTPT) defined in the text (lines 184 and 197) differs from the acronym used in Figures 2 and 3 (OPT-PNT), assuming the two have the same meaning like I believe they do. Corrected to OPT-PNT everywhere.

- Table T2: Column 2, Row 3 has an extra "tab" space that should be deleted. Corrected

- Notation for the equation in section 1.3 of the supplementary information should be made consistent. Most of the terms use forward slashes but "Cost(P|N)" uses a vertical bar in the text. Corrected

**Anonymous Referee #2, 18 Jan 2024**

**General comments:**

One key point: In the history of looking at "damage" (specifically, reflections and noise in radar data caused by crevassed zones at the edges of ice streams) in ice shelves, the shoe started out on the other foot: Signs of damage that were buried under un-damaged firn (specifically at Kamb Ice Stream, then called ice-stream C), and their extensions out on the Ross Ice Shelf were of great significance in telling the story of past change in the flow conditions. In this case, the change was the "shut down" of an ice stream (which has less press value in today's world, but which needs to be studied, especially if there is ever going to be hope that ice-stream flow at Thwaites or Pine Island Glacier will self-limit). It might be worth touching on this point in the introduction. I believe that the original literature on it (from the late 1970's and early 1980's) is easy to find.

We have added a short paragraph in the text (line 47-48): "**Historically, signs of damage concealed beneath undamaged firn in the margins of ice streams have played a crucial role in past changes in ice flow conditions, i.e., the shut-down of Ice stream C in Antarctica, (Robin, 1975, Shabtaie and Bentley, 1987, Retzlaff and Bentley, 1993, Retzlaff et al.,1993, Smith et al., 2002).** "

Generally speaking: the word on the "street" (I got this from an editor's meeting of a related glaciological journal) is that use of acronyms can be an impediment to getting papers to be cited. In some ways, I think this is intuitively obvious; but apparently it is also a result of doing careful quantitative analysis with specific metrics for measuring acronyms and citations. This paper does not have a lot of acronyms, however, I still wonder if the paper would be easier to read (and thus more likely to be cited) if acronyms were minimized. The

ones that I had to struggle with were: CNN, NeRD (that one appears to be a kind of subtle joke, which I like, as the word NeRD in English refers to a "smart" but slightly "dull" person), MOA, ROC, FPR, TPR, AUC, OPTPT, … This is an online journal, hence there is no cost in paper to write out the words in full. I think that the authors should consider this. The authors might additionally find it works more simply to assign actual variables to elements that are now a kind of hybrid acronym, for example: AUC-mean2009. These long, strung out variable names that incorporate an acronym make the reading of the paper a bit harder. With harder reading, there is then the possibility of fewer citations.

While we acknowledge the reviewer's suggestion to expand certain acronyms, we have, however, opted to retain the current format. This decision is driven by the frequent repetition of these acronyms throughout the text and their substantial length when written in full—consider NeRD (NormalisEd Radon transform Damage detection), for instance. We trust that readers will still be able to comprehend the text and interpret the results effectively.

I notice that the analysis makes a distinction between 2019 and 2020 velocities in the AUC for Pine Island Ice Shelf (and different years for different regions). What specifically (as a reminder) is changing? Is it the detected crevassing fields or is it the velocity field?

For Pine Island Ice Shelf, we use the method developed by Izeboud et al. in 2023, which classifies crevasses adopting a composite satellite imagery — here, we use the one from December to February of 2019-2020. To ensure optimal alignment with the crevasse map corresponding to this specific timeframe, we employ two distinct velocity datasets from the years 2019 and 2020 and perform two ROC analyses.

**Specific comments:**

Abstract: "Wealth of research"? I'm not sure it adds precision to use the term wealth as a qualifier. Maybe another word would be more appropriate. "Wealth" appears as the first word in the discussion as well.

Response: We have changed it to " An abundance of studies" for the discussion (line 332).

A minor point: I see that variables that appear in the text are italicized (as they should be), however, this italicization needs to be checked. For example, x- and y- around line 115. Ditto for the R that appears near there. A good double check just for this would be useful.

Response: Thanks, it was checked.

L-curve method (here the L should not be italicized, as I think that the "L" denotes a shape more than a variable). Also, I've never heard of this method before, so I wonder if the reference to it should appear right away. Also, I would find it helpful to possibly say in a few sentences how a user would "walk through" a problem following the L-curve method.

Response: Reference was moved earlier in text; additional text (here in bold) was added to make it easier to understand; please refer to lines 133-144.

 "To avoid the inverse solution either being shaped by the given prior or overfit observations, we adopted the L-curve method (Calvetti et al., 2002) which graphically visualises, in a log-log scale, the relationship between the norm of the regularised solution and the norm of the residual error. **This technique consists of performing multiple**

**simulations in which different regularisation magnitudes are tested and compared to their reciprocal norm residuals.** The distribution of these points follows an L-shape curve and the point where the horizontal and vertical branches converge is the L-corner. **This point corresponds to the point of maximum curvature and represents a solution where the "perturbation" errors and the "regularisation" errors are well balanced. Since the regularisation equation that we solve in our inversions adopts two regularisation parameters, $\gamma_a$ and $\gamma_s$, we systematically have to assess both quantities for both regularisation parameters. Thus, the method performs two sets of simulations: for each set, one of the two regularisation parameters is allowed to evolve by several orders of magnitude, while the other is kept fixed, and the L-corner value is found. For our simulations, we found an ideal L-corner at $\gamma a = 1$ and $\gamma s = 25000$.** Selecting a smaller value for $\gamma_s$ will not affect the calculated velocity distribution significantly, hence any further variation in A resulting from selecting a smaller value is not supported by any corresponding variations in observed velocities. **On the other hand, selecting very large values of $\gamma s$ (> 109) will cause the solution of $A$ to be spatially uniform (we refer to Figure S5 in the Supplement material)."**

Something to check: numbers in the text sometimes appear in scientific notation (where there is a "times ten to the power of something") and sometimes in digital computer notation or floating point notation, e.g., 1.3e-4... Journal style should be checked. I suppose it would be a bit pedantic to say so, since nobody even thinks about this any more: but it would be cool if every now and then people would report whether they are doing single or double precision computations (I don't suppose there are single precision computations any more, but what the heck, I might as well bring it up).

Response: Many thanks, it was checked

Figure 4: I note that the vertical axes have a notation that is " #10^n " Is this standard notation (I usually see "x" replace the "#")? Also, would it be better to have the scale (ten to the power of) in the axis label rather than perched on top of the axis frame?

Response: Corrected

This one is not an essential comment, and is motivated by the fact that Chris Borstad, one of the pioneers of damage mechanics in glaciology passed away in November of 2023. I see that Copernicus no longer provides an "acknowledgment" section in its articles where a dedication (if the authors were to want to make one) would normally appear. Instead, I see that Copernicus prefers to replace the general acknowledgement section with specific (and seemingly less noble) sections like "Funding". This comment is not a criticism, just something that came to my mind (and heart).

Response: If the Journal allows it, we are happy to make a dedication, in the funding section. Thanks for your kind suggestion.

**"This manuscript is respectfully dedicated to the memory of Chris Borstad, a distinguished pioneer in glaciology and damage mechanics, who passed away in November 2023. His significant contributions have greatly influenced our understanding of this field and serve as a source of inspiration that continues to shape and guide the trajectory of our work."**

**Adrien Gilbert, 9th Feb 2024**

This study examines whether the surface crevasse field observed from remote sensing is related to the flow rate factor derived from inverse methods constrained by surface velocity. A poor relationship is found, suggesting that surface crevasse observations may not be a good proxy for quantifying ice damage affecting ice rheology over a thickness relevant to ice flow dynamics.

The study addresses an important topic, as observational constraints on ice damage are critical for assessing ice shelf dynamics and stability in ice sheet models. The methodology is rigorous and clearly described, and the results are convincing and well presented. I recommend the paper for publication in The Cryosphere after major revisions.

**General Comments**

My main concern is that the inferred flow rate factor from surface velocity is consistently presented in the manuscript as the truth that the observed crevasse field should match. I think this way of presenting the study is not fair as it could be turn differently. For example, the crevasse field could be presented as the truth that the inferred flow rate should match, and one could conclude that the model does not capture the effective viscosity and stress field well due to its simplification, inappropriate physics or inaccurate ice shelf three dimensional geometry. The manuscript clearly lacks of a discussion about the model assumption and the reliability of the inferred flow rate factor. Even if the model is strongly constrained by surface velocity observations, it does not sound right to question the utility of crevasse field observations without even mentioning that the weak relationship could be due to the model lacking the relevant physics. Are we sure that the inferred flow rate factor is not affected by neglecting the elastic stress field or the depth-dependent variation of the flow rate factor or other things? The whole study could also conclude that the SSA does not capture the stress field of the ice shelf well, because the inferred value of the flow rate factor is weakly related to the observed damaged areas. This would give the opposite message to the community ...

I suggest that the authors should also consider the case where the SSA approximation is the cause of the discrepancy and provide strong arguments if they think this cannot be the case.

Response:

The reviewer argues that surface crevasses must impact ice rheology, as estimated by state-of-the-art ice flow models.  The main research question addressed in this study is whether this is indeed the case and whether satellite-derived "damage" maps can be used to constrain inverse problems in ice-flow models that infer the ice viscosity from surface velocities. Here, we used classification analyses and found a limited match between modelled and satellite-derived damage. The reviewer questions whether the discrepancy found between these two products is due to the model lacking physics, either by the use of a depth-integrated approximation or by neglecting the elastic component of deformation.

Since we are studying ice shelves, it is conventional in ice flow modelling to employ the SSA (Shallow Shelf Approximation) equations. This approximation stems from the relatively thin nature of ice shelves compared to their horizontal extents and it assumes that horizontal stresses within the ice shelf are primarily controlled by the horizontal

velocity gradients, neglecting vertical shear stresses and variations in ice thickness. There are certain regions where indeed the SSA may not behave well / break down, i.e., in areas close to the grounding line, in regions where there's a drastic change in slope or topography, close to pinning points and ice rises, due to the presence of high vertical shear. In this work, we have investigated the depth-integrated ice viscosity for the Filchner Ronne and Pine Island Ice Shelves and have ignored for the ROC analysis all regions of "A" that were within 5 km of the grounding line. This was not mentioned in the text earlier and we have added it at lines 171-175.

The advantage of adopting the SSA equations is that they provide a depth-integrated solution, so whenever we invert for "A", by fitting horizontal surface velocities, we obtain a depth-integrated ice rate factor that depicts the depth-integrated properties of the ice for that moment in time. This allows us to identify weakened ice throughout the ice shelf thickness, even where there is no surface expression of ice damage or weakening. In this work, we showed that most of the surface crevasses mapped by satellite imagery are shallow features that do not have an impact on the depth-integrated ice viscosity and ice flow.

The reviewer questions the ability of the model to accurately depict areas of damage due to a lack of physics and the use of SSA. In response, we direct the reviewer to a study by DeRydt et al., 2019, where the same model effectively identified Chasm 1 and Halloween crack on the Brunt ice shelf. In that study, the model adopted the same setup as this work, employing the SSA approximation and inversion methods to estimate the spatial distribution of "A". They examined nine different configurations of the ice shelf between 1999 and 2017, based on snapshot observations of surface velocity, ice thickness, and ice shelf extent, investigated the timing and location of rift formation and looked at mechanical changes in the ice shelf by analyzing spatial maps of principal stress magnitude and direction. Figure A2 in the Appendix of the article illustrates the distribution of "A" before and after rift formation, showing higher values of "A" (weakening of the ice) along the trajectory of active rifts, Chasm 1 and Halloween crack, and at the hinge zone immediately downstream of the grounding line. These areas of soft ice accommodate the high strain rates or discontinuities in flow speed in those locations. Bands of stiffer (colder) ice are also seen along flow lines from the grounding line to the ice front and have previously been identified via ground penetrating radar as bands of meteoric ice originated upstream of the GL, in contrast to the surrounding areas that predominantly consist of (warmer) marine ice (King et al., 2018). These results overall suggested not only that the model accurately represented active rifts as areas of weakness but also that the distribution of "A" was meaningful and further supported by other independent work.

A similar weakening is visible in our study when analyzing the "A" distribution for Pine Island Ice Shelf fitting velocities of November 2019, a moment in time preceding the calving event that took place on February 11[th], 2020. We refer the reviewer to the additional Figure S11 in our Supplementary Material which displays the inverted ice rate factor and observed speed for Pine Island Ice Shelf, in 2019 and 2020. Looking at the 2019 velocity map (S11b), we see a large jump in velocity close to the ice front, where an ice rift is present. This abrupt change in velocity across the rift flanks is detected by our model as an increase in strain rate and an increase in the solution of A (S11 a), thus displaying a transverse region of weakened ice in that exact region where the rift is located, and where the calving event ultimately took place.

In summary, by properly fitting surface velocities, the model can correctly represent the stress distributions and deformation across the shelf and accurately identify rifts and

weakened ice regions. As argued by the reviewer, improvements can always be made by accounting for the full stress tensor; such measures are nevertheless unnecessary given the model's already precise representation of the ice shelf stress and strain distribution.

The second point of the reviewer was the disregard of the elastic stresses. Here, we refrain from investigating these effects, since most of the ice behaves as a viscous fluid, and instances of elastic behavior may arise in limited settings: when a rift propagates in rapid episodic bursts in very short timescales, or at the bending zone region downstream of the grounding line, due to the impact of tides and tidal flexure. In the current context, we find these effects to be irrelevant to the outcomes of our study. In fact, elastic effects can be safely ignored at stresses and strain rates typical of ice shelf flow. We refer the reviewer to Gudmundsson et al., 2007, Figure A1. For very short loading times, the response of the ice is indeed elastic with a modulus equal to the instantaneous Young's modulus of ice. For very long loading times, the ice behaves as a viscous material with an effective viscosity related to englacial temperature and effective stress. Between these two limits, there is a range of loading periods for which the ice behaves elastically but with a Young's modulus related to delayed elastic response (primary creep). Loading periods/strain rates in ice shelves are about 5 orders of magnitude too small for elastic effects to be of relevance. For this reason, we disregard any effects of elastic deformation.

The additional text to address your comment is reported here in bold. We have added a whole paragraph in the classification technique section at lines: 171-175

**"Since we employed the SSA equations to invert for the solution of A, there are certain regions for which the SSA equations may not behave well / break down, i.e., in areas close to the grounding line, in regions where there's a drastic change in slope or topography, close to pinning points and ice rises, due to the presence of high vertical shear. To ensure reliable results from our ROC analysis we have excluded all areas of A that were found within a 5 km radius of the grounding line."**

And in the discussion section at lines: 366-376

**"The observed discrepancy between modelled damaged and remotely sensed crevasse maps is not to be attributed to deficiencies in the physics incorporated within the model: the use of a depth-integrated approximation or the omission of the elastic component of deformation. Our methodology, which leverages SSA equations and inversion techniques, has proven effective in identifying rift formation and pinpointing their locations (DeRydt et al., 2019), yielding an inverted solution that not only accurately located areas of weakening but also faithfully replicated independent analyses of ice rheology (King et al., 2018). Our study abstained from investigating elastic effects, firstly as the predominant behaviour of ice resembles that of a viscous fluid, and instances of elastic behavior arise in limited settings; secondly, as elastic effects can be safely disregarded at stresses and strain rates typical of ice shelf flow — given that loading periods and strain rates in ice shelves are approximately five orders of magnitude too small for elastic effects to be significant (Gudmundsson et al., 2007). The limited agreement between the two damage products suggests that the majority of surface crevasses identified through satellite imagery are shallow features that do not exert a discernible impact on the depth-integrated ice viscosity and ice flow. "**

**Specific comments**

Line 31: what is fracture data? You could clarify.

Response: Surawy-Stepney et al., (2023) proposed this hypothesis in their study; as we are testing the same hypothesis, we use the same wording and terminology. We have added in line 30 in brackets: "**(remotely sensed maps of ice fractures)**"

Line 72: You could elaborate here about why a classification problem rather than a simple correlation coefficient.

Response: Classification analyses such as ROC provide a more comprehensive assessment of model performance, as they can handle imbalanced datasets more effectively than correlation coefficients, which might be biased towards the dominant class. Moreover, the ROC analysis provides model performance across various threshold levels, giving insights into how changes in the threshold affect the trade-off between true positive rate and false positive rate. These advantages are extensively detailed throughout the manuscript, particularly in the methodology and in the Classification Techniques section. We have added a section here at line 75 : **"Classification analyses offer a more comprehensive assessment of model performance compared to correlation coefficients, since they can handle imbalanced datasets better and provide insights into the model's performance across different threshold levels."**

Line 72: Why not opposite ? I though it would be more logical to treat the damage as a predictor for ice rate factor since the damage map are the observations. Does it change something ?

Response: This paragraph adopts the terminology frequently used in ROC analyses. The text says: *"*We treat our model inverted ice rate factor as a predictor for damage and quantify how often it corresponds to crevassed areas (true positives) as against to how often crevasses are incorrectly predicted from areas of damage (false positive)." To make it clear, we added further text (line 77): **"The crevasse products obtained by remote sensing/machine learning techniques represent the true observations to be classified (that is, to match). The predictor variable is the variable used to make a prediction."**

Line 156: The meaning of "classify areas of damage" is not clear to me. Is there different type of damage expected from remote sensing observations ? Or do you mean classify between damaged and not damaged area ?

Response: The text read:

"If there is a strong link between the inverted ice rate factor $A$ and the damage maps obtained from remote sensing methods, we would expect to be able to use the inverted $A$ field to classify areas of damage as identified through the remote sensing methods."

Here again we use the classification terminology: "classify" . We have added further text to make it clear (line 170): **predict crevasses where satellite maps detected crevasses (True Positive).**

Line 215: Increase font size of legends in the two crevasse maps

Response: Done

Line 215: why not using recommended value from Cuffey and Paterson (2010) rather than value coming from one specific study?

Response: It corresponds to the same value. We have added Cuffey and Paterson (2010) as a citation.

Line 332: how ? Using recommended Arrhenius law from Cuffey and Paterson 2010 ?

Response: We adopt the same approach of Spring and Morland, 1983. We have added it as a citation.